# The Three Pillars of COVID-19 Convalescent Plasma Therapy

**DOI:** 10.3390/life11040354

**Published:** 2021-04-18

**Authors:** Massimo Franchini, Giancarlo Maria Liumbruno, Giorgio Piacentini, Claudia Glingani, Marco Zaffanello

**Affiliations:** 1Department of Hematology and Transfusion Medicine, Carlo Poma Hospital, 46100 Mantova, Italy; giancarlo@liumbruno.it (G.M.L.); claudia.glingani@asst-mantova.it (C.G.); 2Department of Surgical Sciences, Dentistry, Gynecology and Pediatrics, University of Verona, I-37126 Verona, Italy; giorgio.piacentini@univr.it (G.P.); marco.zaffanello@univr.it (M.Z.)

**Keywords:** convalescent plasma, COVID-19, hyperimmune plasma, SARS-CoV-2, therapy

## Abstract

The new severe acute respiratory syndrome coronavirus 2 (SARS-CoV-2) infection has spread rapidly around the world in the last year causing the coronavirus disease 2019 (COVID-19), which still is a severe threat for public health. The therapeutic management of COVID-19 is challenging as, up until now, no specific and efficient pharmacological therapy has been validated. Translating the experience from previous viral epidemics, passive immunotherapy by means of plasma from individuals recovered from COVID-19 has been intensively investigated since the beginning of the pandemic. In this narrative review, we critically analyze the three factors, named “pillars”, that play a key role in determining the clinical effectiveness of this biologic therapy: the convalescent plasma, the disease (COVID-19), and the patients.

## 1. Introduction

Coronavirus disease 2019 (COVID-19) is a still ongoing pandemic caused by the severe acute respiratory syndrome coronavirus 2 (SARS-CoV-2), initially discovered in Wuhan, China, at the end of 2019 and quickly disseminating all over the world [1]. As of 1 March 2021, the infection had already affected approximately 120 million people and caused nearly 2,500,000 deaths worldwide, and the rates are still increasing, according to World Health Organization (WHO) bulletins [2]. Unfortunately, current treatment options are limited. Corticosteroid therapy has been shown to significantly reduce 28-day all-cause mortality in severely ill COVID-19 patients compared with usual care or placebo (odds ratio (OR) 0.66; 95% confidence interval (CI), 0.53−0.82]; *p* < 0.001) [3]. Among the few available therapeutic options, the use of plasma from COVID-19-recovered donors has been the object of an intense research from investigators during the last 12 months [4,5]. Convalescent plasma (CP), containing neutralizing anti-viral antibodies, is a form of passive immunotherapy that has been used for the treatment and prevention of infectious diseases for more than 100 years [6]. CP was successfully used in the treatment of severe acute respiratory syndrome (SARS) in 2002, Middle East respiratory syndrome (MERS) in 2012, and the 2019 H1N1 pandemic [6]. A meta-analysis of 32 SARS and severe influenza studies showed that CP treatment was associated with a significant reduction of mortality (pooled OR 0.25; 95% CI, 0.14−0.45) [7]. 

Following the negative results from certain randomized controlled trials (RCTs) [8,9], some scientists have claimed that CP does not work against COVID-19. However, in our opinion, the issue is wrongly posed. Indeed, assuming that CP containing high-titer anti-COVID-19 neutralizing antibodies is effective in suppressing viral replication [10,11], the correct question these scientists should pose is “why does CP not work in our study?”. This is probably because there are some critical issues in the study design. However, to respond correctly to this question, one must take a step back, separately analyzing the three key factors determining the clinical effects of CP therapy. We have called them “pillars”, translating our transfusion experience in the patient blood management (PBM) setting. The three “pillars” of CP therapy are the treatment (hyperimmune plasma), the disease (COVID-19), and the patients (Figure 1). This narrative review aims at elucidating the clinical role of these three CP-related factors.

## 2. Search Strategy

As a search literature strategy, the Medline, PubMed, and Google Scholar electronic databases were searched for publications on the clinical efficacy of CP in COVID-19 patients between January 2020 and March 2021, using English language as a restriction. The Medical Subject Heading and key words used were “convalescent plasma”, “hyperimmune plasma”, “hyperimmune serum”, “therapy”, “SARS-CoV-2”, “COVID-19”, “coronavirus”, “neutralizing antibodies”, “mortality”, “safety”, “clinical trials”, and “treatment”. We also screened the reference lists of the most relevant review articles and systematic reviews and meta-analyses for additional studies not captured in our initial literature search.

## 3. The Convalescent Plasma

The first initiative that the virologists took at the beginning of COVID-19 outbreak was that of testing the plasma from COVID-19 patients with a cell culture infected from SARS-CoV-2 [4]. The ability of that plasma to prevent cellular infection by the virus documented the presence of neutralizing anti-SARS-CoV-2 antibodies and paved the way for the production of CP from recovered individuals to treat COVID-19 patients. Although various mechanisms of action of CP have been proposed [10], hyperimmune plasma works mainly because it has anti-SARS-CoV-2 antibodies capable of preventing the virus from entering cells, infecting them, and subsequently replicating. It follows that the antiviral activity of the CP is linked to the number of antibodies present—the more the neutralizing antibodies, the more effective the plasma will be in blocking viral replication [11]. The importance of high-titer CP clearly emerges from the analysis of the RCT by Libster and colleagues [12], wherein a dose-dependent effect size was detected—recipients of CP at a titer higher than the median concentration showed a relative risk reduction of 73% in disease progression, which reduced to only 31% when CP was given at a titer below the median concentration. This paradigm has been demonstrated, albeit indirectly, by the Indian RCT study in which it was found that CP with a low titer of anti-SARS-CoV-2 neutralizing antibodies is not effective against COVID-19 [8]. Therefore, assuming that the neutralizing potency of CP is the most critical factor in determining its clinical effectiveness [13], it follows that its accurate evaluation is mandatory to obtain a high-quality CP product. The current gold standard to assess viral neutralization by CP is the plaque reduction neutralization test (PRNT), which measures the ability of neutralizing antibodies to prevent infection in vitro calculated as a reduction in the formation of plaques [14]. This assay utilizes live SARS-CoV-2 virus and, hence, requires a biosafety level 3 (BSL-3) facility. In addition, it is time-consuming (5–7 days) and non-standardized [15]. To overcome these drawbacks, several investigators have developed a number of immunoassays, such as enzyme-linked (ELISA) and chemiluminescent immunoassays (CLIA), as surrogate of PRNT [16]. Although such tests showed broad degrees of variability pertaining to the sensitivity and specificity when compared with PRNT, their diagnostic accuracy, which combines sensitivity and specificity and is represented by the area under the curve (AUC) in the non-parametric receiver operating characteristic (ROC) curve method, is often suboptimal [17,18,19]. Therefore, serologic immunoassays are useful for the initial screening of candidate CP donors according to pre-specified cut-offs of correspondence with PRNT, which is, however, mandatory to exactly assess the anti-viral potency of CP before its release for clinical use [15]. On the other hand, the continuous qualitative development of commercial serological high throughput SARS-CoV-2 assays lets us foresee a future use in the determination of the neutralizing potency of CP [20]. High-throughput single B cell sequencing technologies have been recently developed to rapidly identify the most potent anti-SARS-CoV-2 neutralizing antibodies [21]. 

## 4. The Disease

Another critical issue for assessing the CP efficacy is its timing of infusion in COVID-19 patients. At the beginning of the pandemic, CP was transfused in more critical patients, such as those hospitalized in intensive care units (ICU) and under mechanical invasive ventilation. For these patients, indeed, there was at that time no therapeutic chance. The RCT by Li and colleagues [22], published as early as August 2020, did not demonstrate a statistically significant difference in 28-day mortality between the CP-treated and standard treatment groups. However, stratifying patients by disease severity (severe or life-threatening COVID-19), the researchers observed a statistically significant difference in time to clinical improvement within a 28-day period in the group treated with hyperimmune plasma. Therefore, this study highlighted that CP must be administered at an early stage of the disease and not in an advanced phase in order to have the maximum effect. This observation is also plausible with the high replication kinetics of SARS-CoV-2 and with the mechanism of action of hyperimmune plasma, considering that the blocking of viral replication during the initial phase of COVID-19 is also able to prevent the activation of inflammatory and coagulative cascade, often irreversible, which instead characterizes the advanced stage of the disease where, unfortunately, the viral anti-replicative activity of CP is, at that stage, ineffective. In other words, it is essential to prevent the COVID-19 from progressing through the early administration of hyperimmune plasma and the patient from undergoing invasive mechanical ventilation and being transferred to the ICU. In fact, once intubated and in the ICU, the patient is at a very high risk of mortality linked above all to superinfections favored by immunosuppression caused by virus-associated lymphopenia [22]. Paradoxically, a consistent number of critically ill COVID-19 patients die without having SARS-CoV-2 infection (negative search for SARS-CoV-2 nucleic acid), but for its deleterious consequences. Another RCT conducted by Libster and colleagues [12] enrolled older individuals with COVID-19 who were identified in the outpatient setting within 48 h of symptom onset. The patients who were given CP within 72 h of symptom onset had a 48% reduced risk of progression to severe respiratory disease. The benefit of administering CP early in the disease course is further corroborated by data from observational studies. An analysis of a cohort of 3082 patients in the U.S. Expanded Access Program (EAP) found that high-titer CP given less than 72 h after hospital admission conferred a survival benefit when compared to those receiving CP later in their hospital stay [23]. A matched propensity score study published by Salazar and colleagues found the greatest effect when patients were given CP within 44 h of hospital admission [24]. Thus, thanks to these and other similar studies [25], nowadays we are aware that CP must be administered early, possibly within 72 h from symptom onset [26]. Other RCTs [27,28] did not find clinical benefit from later CP administration. On the basis of the newer literature evidence, the U.S. Food and Drug Administration (FDA) recently revised the Emergency Use Authorization (EUA) of COVID-19 CP, authorizing its use at high titer for the treatment of hospitalized COVID-19 patients early in the course of disease and those hospitalized with impaired humoral immunity [29]. Finally, clinical studies are needed to verify the possible resistance of viral mutations to CP therapy observed by in vitro studies [30].

## 5. The Patients

The identification of the demographic and clinical characteristics of COVID-19 patients who could benefit most from CP is still a controversial issue and a matter of discussion among researchers. On a rational basis, CP should have it maximum efficacy in patients with an impaired humoral response against COVID-19 [26]. Indeed, there are several reports on CP clinical benefit in such patients, including those with solid and hematologic cancers and with acquired and congenital immune deficiencies [31]. In particular, a recent systematic review reported positive outcomes following CP infusion in 14 COVID-19 pediatric patients, wherein five of them suffered from congenital immuno-deficiency or onco-hematologic disorders [32]. There are also some reports documenting the clinical efficacy of CP in COVID-19-complicated pregnancies, where an immuno-deficient state can be considered physiologic [33]. 

In addition, the previously mentioned RCT by Libster and colleagues [12] showed that mildly SARS-CoV-2-infected older adults benefit from CP infusion to prevent the disease progression. These data have been corroborated by the results of a recent single-center study published by our group in 22 elderly COVID-19 patients [34]—the early administration of high-titer CP was safe and led to a rapid viral clearance, halting the progression of COVID-19 and thus resulting in a survival benefit. Aging, indeed, is another example of a physiologic process characterized by an impaired function of the immune system [35]. However, does this mean that CP in younger individuals is not indicated? Absolutely not. In addition to the antibody-based inhibition of viral replication, there are other important actions exerted by CP useful in fighting COVID-19, such as the anti-inflammatory and immunomodulation activities (Figure 2) [10]. Lastly, the recent reporting of the anti-thrombotic activity of albumin, the most abundant plasma protein, sheds new light on the mechanisms of action of CP [36]. One proposal could be that of selecting among younger COVID-19 patients those who might have the greatest benefit from CP passive immunotherapy, such as those with insufficient anti-SARS-CoV-2 neutralizing antibody levels. However, although the Dutch RCT [37] has suggested the inefficacy of CP in patients with high baseline SARS-CoV-2 neutralizing antibody titers, it is difficult to establish a priori a patient antibody titer threshold below which the CP transfusion is indicated. Unfortunately, this type of information is not available from already published studies and very few ongoing trials are exploring this issue. In addition, focusing only on this aspect related to CP efficacy is not completely correct because, as previously mentioned, the mechanisms of action of CP certainly include other factors not related only to the viral neutralizing effect of anti-SARS-CoV-2 antibodies [10]. Finally, it must be outlined that there are no published or ongoing dose-finding studies—the modality of CP administration greatly varies from study to study, ranging from 200 mL to 600 mL per dose, and can be usually repeatable for up to two or three times at 24−72-h intervals. Furthermore, the clinical response to a transfused CP unit might depend on patients’ body mass index (BMI). Therefore, adequately powered trials must be designed not only to assess the best patients’ neutralizing antibody threshold for CP transfusion but also to optimize CP dose and treatment modality.

## 6. Conclusions

After a careful analysis of the published literature studies regarding the main factors implicated in CP clinical efficacy, i.e., the CP product and the characteristics of COVID-19 patients and of the disease, we realized that there are still some gray zones and unanswered questions in this area. Nevertheless, the great majority of available literature agrees on the efficacy of CP in blocking promptly SARS-CoV-2 viral replication. That said, it is unlikely that this biological activity by CP has no clinical consequences—it is only a question of knowing how to identify them. The current literature evidence is clearly summarized by the recent interim recommendations from the AABB [27]. Besides the safety of CP, which has been considered comparable to standard non-hyperimmune plasma, researchers have pointed out high-titer and transfusion as close to symptom onset as possible as the main predictors of effectiveness of CP. Thus, considering the results from the literature supporting the efficacy of early treatment of COVID-19 patients, an outpatient CP transfusion approach in order to treat earlier COVID-19 patients preventing hospitalization (with all the risks associated with immobilization and hospital co-infections) could be reasonable. This argument is, however, not speculative but based on the clinical practice observation that CP transfused to elderly patients with moderate-severe COVID-19 living in long-term care facilities is safe and effective in blocking disease progression and reducing the mortality risk by 65% [33]. Unfortunately, in Italy (and other European countries), CP can be administered only in hospitals within experimental trials or for compassionate use, and this greatly limits its widespread and potentially more appropriate clinical use.

Since we would have to live with COVID-19 for a few more months before reaching mass immunity with vaccines, we hope that on the basis of further clinical evidence also in Europe, as in the USA, governments will take the necessary actions in order to favor the emergency use of CP even outside the hospital setting. In parallel, dose-finding trials aimed at better tailoring CP therapy to patients’ characteristics are welcomed.

## Figures and Tables

**Figure 1 life-11-00354-f001:**
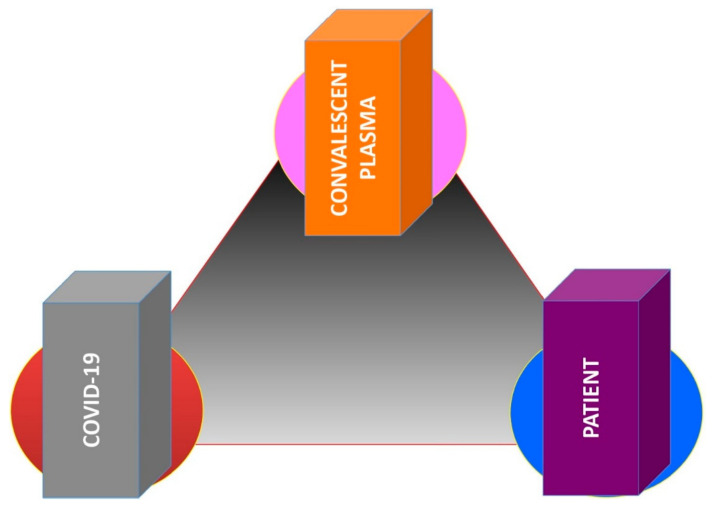
The three pillars of anti-COVID-19 convalescent plasma therapy.

**Figure 2 life-11-00354-f002:**
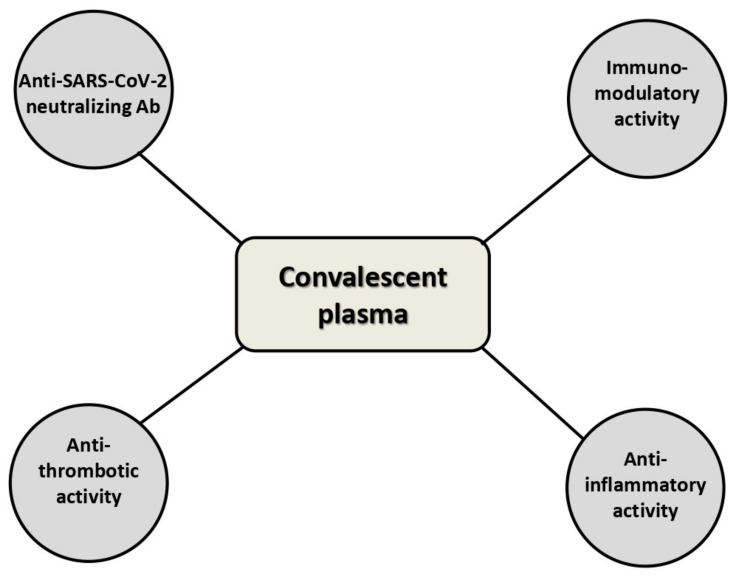
Mechanisms of action of anti-COVID-19 convalescent plasma.

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
