# Peer review of "The Three Pillars of COVID-19 Convalescent Plasma Therapy"

_life, 2021, doi:10.3390/life11040354_

Round 1
Reviewer 1 Report
Overall, this is an important discussion and topic for treatment of COVID-19. Additional background information on the disease progression in the patient and possible treatment times may be beneficial. The text contains many opinion statements from the authors and it is recommended these statements be re-worded to reflect actual findings or include references as appropriate.
Lines 42-47: These are opinionated statements with no references to confirm the conclusions made. These statements should be re-worded to reflect data or observations from the authors without inferences to other studies.
Lines 64-66: Please include a reference to offer evidence for this statement.
Line 67: Was the word 'serum' inadvertently used here in place of plasma? If correct, can this statement be reworded for clarification?
Line 76: How was the titer/median concentration determined? Was this an ELISA, a cell-based neutralizing assay, etc.? What are the units for the 'titer'?
Lines 81-82: This is an opinion. Please provide references or reword.
Lines 99-101: Please remove opinion statements "we think" and general terminology such as 'probably'.
Have the authors considered the microneutralization assay similar to the PRNT using live virus, which can be performed as a 2-day assay using an in-situ ELISA readout?
Lines 111-112: Please reword general statements such as 'helped us'.
Lines 127-131: Please include references for these statements.
Line 143: Replace the statement 'negative results' as it reflects an opinion of the authors. If this is referencing 'negative testing results', please clarify.
Line 147: Suggest including information on titer.
Lines 167-169: Opinion statements without references should be removed.
Line 182, 187/188: Please revise opinion statements, such as 'not completely correct because...' and 'transfusing a CP unit.... is not the same!'.
Lines 212-215: Revise opinion statements and verbiage.
Lines 217-244: Please remove sections that do not apply to the text.
Author Response
Answers to Reviewer: 1
Lines 42-47: we have added the references, as suggested.
Lines 64-66: done.
Lines 67: done.
Line 76: The authors used used the SARS-CoV-2 Spike S1-RBD IgG ELISA detection kit (GenScript) and the SARS-CoV-2 surrogate virus neutralization test kit (GenScript). High titer was defined as > 1:3200.
Lines 81-82: We have deleted this sentence.
Lines 99-101: done. We did not considered micro neutralization assay (similar to PRNT) as this discussion is focused on high throughput SARS-CoV-2 assays and their accuracy to detect neutralizing antibodies. We have deleted reference 20 and renumbered the others accordingly.
Lines 111-112: done.
Lines 127-131: done.
Line 143: done.
Line 147. The definition of high-titer convalescent plasma in the FDA document depends on the type of assay used. It is therefore not possible to summarize all these data in the manuscript.
Lines 167-169: We have added a reference.
Lines 182, 187/188: done.
Lines 212-215: This message is very important for us. It is the most important message of our study.
Lines 217-244: done.
Reviewer 2 Report
Franchini et al. wrote a short, timely review article on Covid19 convalescent plasma therapy. They focused on three factors which they called pillars, namely, the convalescent plasma, the disease and the patients, as they determine the clinical outcome. The information on the COVID19 is now very rapidly updated and the authors need to verify any recent FDA updates or clinical results but not covered here as available if any. In the convalescent plasma section, authors need to comment on antibody therapies from convalescent patients Ex. https://pubmed.ncbi.nlm.nih.gov/32425270/
The disease section, mentioning about new variants and their possible impact of CP therapy may be added. The patients section is fine about the effect of aging and doses. The sex factor can be added as men vs women from recent studies.
A line about current vaccines should be added. In the title "Anti" can be deleted.
Author Response
Reviewer: 2
We have already reported the updated FDA document (reference no. 28).
We have added (reference number 21) and commented the suggested reference. We have renumbered the other references accordingly.
New variants and impact on CP therapy (disease section): we have added a comment and a reference (number 29). We have renumbered the other references accordingly.
Sex factors are involved in CP donors and collection, which are not the object of this study, which is focused on patients.
COVID-19 Vaccines: This is not the object of this review.
Title: done.
Reviewer 3 Report
The Authors of the paper "The three pillars of anti-COVID-19 convalescent plasma" present a review of crucial factors which can influence the clinical effectiveness of CP. Despite negative results from randomized trials, the Authors suggest that such therapy could be efficient under certain conditions and search for the main predictors of its efficacy. All of the three pillars are critically analyzed, and among others, the role of administration of CP at an early stage of the disease is highlighted. In the Introduction Authors state that current treatment options in COVID-19 are very limited and mention only corticosteroid therapy, whereas the role of antiviral treatment with remdesivir is established, I suggest the Authors include this issue in the manuscript. I also suggest the Authors consider citing real-world data relevant to the paper (Moniuszko-Malinowska A, et al. Convalescent Plasma Transfusion for the Treatment of COVID-19-Experience from Poland: A Multicenter Study. J Clin Med. 2020 Dec 24;10(1):28. doi: 10.3390/jcm10010028. PMID: 33374333; PMCID: PMC7795721). I believe that this will enhance its value.
Author Response
Answers to Reviewer 3.
The role of Remdesivir in COVID-19 is not definitely established.
We have added this important reference (no. 25) and renumbered the others accordingly.
Round 2
Reviewer 1 Report
Thank you for adequately addressing comments and addition of references.